# New Evaluation Metrics Capture Quality Degradation due to LLM Watermarking

**Karanpartap Singh**                                    *karanps@stanford.edu*
*Department of Electrical Engineering*
*Stanford University*

**James Zou**                                            *jamesz@stanford.edu*
*Department of Biomedical Data Science*
*Stanford University*

**Reviewed on OpenReview:** *https://openreview.net/forum?id=PuhF0hyDq1*

## Abstract

With the increasing use of large-language models (LLMs) like ChatGPT, watermarking has emerged as a promising approach for tracing machine-generated content. However, research on LLM watermarking often relies on simple perplexity or diversity-based measures to assess the quality of watermarked text, which can mask important limitations in watermarking. Here we introduce two new easy-to-use methods for evaluating watermarking algorithms for LLMs: 1) evaluation by LLM-judger with specific guidelines; and 2) binary classification on text embeddings to distinguish between watermarked and unwatermarked text. We apply these methods to characterize the effectiveness of current watermarking techniques. Our experiments, conducted across various datasets, reveal that current watermarking methods are moderately detectable by even simple classifiers, challenging the notion of watermarking subtlety. We also found, through the LLM judger, that watermarking impacts text quality, especially in degrading the coherence and depth of the response. Our findings underscore the trade-off between watermark robustness and text quality and highlight the importance of having more informative metrics to assess watermarking quality.

## 1 Introduction

The advancement of Large Language Models (LLMs) like GPT-4 and Llama-2 has heralded a new era in natural language processing, offering unprecedented capabilities in generating human-like text (OpenAI, 2023; Touvron et al., 2023; Chowdhery et al., 2022). However, this advancement also brings forth a unique challenge: ensuring the integrity and traceability of machine-generated content (Clark et al., 2021; Mora-Cantallops et al., 2021). These concerns have led to the development of many watermarking techniques for LLMs, aimed at embedding identifiable markers into generated text without compromising its quality or readability (Kirchenbauer et al., 2023a;b; Takezawa et al., 2023; Zhao et al., 2023a; Christ et al., 2023; Yoo et al., 2023; Zhao et al., 2023b).

Watermarking in the context of LLMs is a relatively new and rapidly evolving field. The primary objective is to embed a non-obtrusive, detectable marker within the text generated by these models, enabling the identification of the source model and potentially deterring misuse such as plagiarism or misinformation. Recent advances have introduced sophisticated techniques aimed at embedding watermarks seamlessly into the language model's output through black-box approaches, ensuring minimal distortion or impact to generation quality without access to the original model's weights. Low distortion is desirable because it shows that watermarking does not introduce significant side effects. However, the effectiveness and subtlety of these methods remain under scrutiny (Tang et al., 2023). Key concerns include the detectability of these watermarks by third parties, their potential to degrade text quality, and the challenge of maintaining the

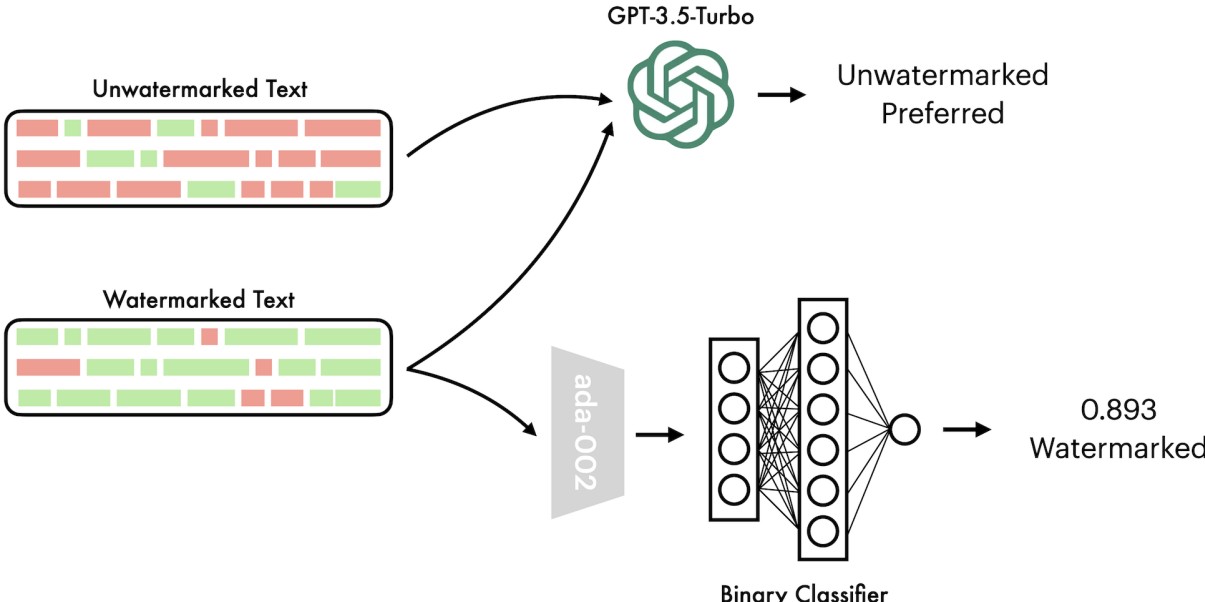

Figure 1: Can watermarked outputs from large language models be distinguished with a black-box approach? We answer this question through two new methods for evaluating LLM watermarks, showing that independent classifiers and judgers with no prior knowledge of watermarking algorithms prefer or can effectively classify watermarked outputs.

watermark's integrity across different contexts and content types. While robustness, or the detectability of a watermark by the intended, knowledgable party using the proper algorithm, is a positive feature, detectability by an uninformed third party with no knowledge of the watermarking secret key or algorithm indicates a lack of subtlety and potential for security risks.

Prior work has used metrics such as perplexity (Kirchenbauer et al., 2023a), n-gram log diversity (Kirchenbauer et al., 2023b), or BLEU scores for machine-translation tests (Takezawa et al., 2023) to evaluate the quality impact of watermarking. However, these metrics fail to capture factors such as the semantic coherence or contextual relevance of the generated text. Additionally, watermarking algorithms may deliver similar results on these metrics to unwatermarked text, while still being discernible by other methods.

In this work, we propose two new benchmarks to assess watermarking algorithms for large language models. We focus our work on determining whether an independent classifier can distinguish a generation as being watermarked without prior knowledge about the watermarking algorithm or associated secret keys. Detectability is a natural way to quantify distortion; the more distorted the generation is compared to unwatermarked text, the easier it would be for a classifier to detect. Through experimentation across various algorithms and datasets, we demonstrate that current watermarking methods can indeed be detected by such classifiers. This finding challenges the prevailing notion of watermark subtlety and calls for a reevaluation of current techniques. By exploring the robustness and quality impact of different watermarking algorithms, we aim to advance the field towards developing more effective and less intrusive watermarking solutions for LLMs.

## 2 Methods

### 2.1 Evaluation Mechanisms

We present two evaluation mechanisms to assess the robustness and quality of the outputs generated by the watermarked models:

1. **Automated Judgement with GPT / GPT-Judger**: We employed GPT-3.5-Turbo with a tailored prompt to act as an impartial judge, inspired by Zheng et al. (2023), and rank generated outputs (watermarked and unwatermarked) on a 1-5 Likert scale for the following factors, selecting by prompting GPT-4 for an appropriate set of criteria for assessing language model generations: relevance to the prompt, depth of detail, clarity of writing, coherence and logical flow, originality and insight, use of specific examples, and accuracy of information. We chose GPT-3.5-Turbo for our primary analyses because of its balance of capabilities and cost/accessibility. However, to assess the impact of the LLM used, we also conducted a comparison study between the detailed prompt with Llama-2-7B-chat, GPT-3.5-Turbo, and GPT-4, as well as a simpler prompt without the categorical ranking system with GPT-3.5-Turbo.

   The prompt instructed the judger to provide specific examples and reasoning for its scoring, as well as a final verdict for which output it preferred overall. Lastly, to account for any positional biases inherent to GPT (Wang et al., 2023a), we randomized the order of the outputs presented to the judger. The full prompts and representative responses are included in Appendix B.

2. **Binary Classifier**: Based on text embeddings obtained using OpenAI's `text-embedding-ada-002` model from the two outputs (watermarked and unwatermarked), we trained a simple multi-layer perceptron (MLP)-based binary classifier, consisting of 961 neurons in 4 layers, to classify a given text as either unwatermarked or watermarked. We also performed a hyperparameter search for each experiment and dataset to ensure the best performance. More details regarding the network and its training are presented in Appendix C. We also tested simple logistic regression on the same embeddings as a classifier, and used k-fold cross-validation random shuffling and 5 folds whenever appropriate.

## 2.2 Datasets

We tested three datasets in this study. For all datasets, a section of text up to 50 words long was spliced from each sample, after which the 7 billion parameter variant of the Llama-2 model was tasked with completing the output (Touvron et al., 2023), both with and without a watermarking layer applied.

1. **LongForm, Validation Set**: we used the Wikipedia subset of the LongForm dataset's validation set, consisting of 251 human-written documents on various topics (Köksal et al., 2023).

2. **C4-RealNewsLike, Validation Set**: A subset of the Common Crawl web crawl corpus, the RealNewsLike dataset contains text extracted from online news articles (Raffel et al., 2019). We used 500 samples from this dataset.

3. **Scientific Papers, Test Set**: A collection of long, structured documents from the arXiv and PubMed open access article repositories (Cohan et al., 2018). We used the abstracts from 252 samples.

## 2.3 Watermarking Techniques

We examined four distinct watermarking techniques, the Soft-Watermark (Kirchenbauer et al., 2023a), Robust Distortion-Free Watermark (Kuditipudi et al., 2023), NS-Watermark (Takezawa et al., 2023), and Unigram Watermark (Zhao et al., 2023a) in this study, though our evaluation metrics could be applied to any watermark using a black-box approach, regardless of its complexity or other attributes. In selecting these techniques, we prioritized methods that were readily reproducible with published codebases. We study the soft-watermark for our primary analysis because of its computational efficiency, popularity, and its similarity to many subsequent methods, such as the NS and Unigram watermarks. Additionally, we also study a very recent distortion-free watermarking technique to illustrate how our metrics can reveal limitations in watermarking methods that explicitly aim to maintain the original quality in the generated text. We report our observed true and false positive rates for each watermarking method in Appendix A.

1. **Soft-Watermarking**: Soft-watermarking, involves the pre-selection of pseudo-random "green" tokens before word generation, promoting their use during sampling (Kirchenbauer et al., 2023a). This watermark can be detected through a statistical test with knowledge of the secret key and token hashing function used to select the green list. This technique was optimized for improved reliability in Kirchenbauer et al., and we use the updated SelfHash scheme in all of our experiments (Kirchenbauer et al., 2023b). The watermarking parameters remain fixed at $(\gamma, \delta) = (0.25, 4.0)$ unless otherwise stated. These parameters were chosen using recommendations from Kirchenbauer et al. (2023a), along with testing on Llama-2-7B to ensure that watermarks with sufficiently high z-scores (z=4) were implanted.

2. **Robust, Distortion-Free Watermarking**: This watermarking technique involves using inverse transform sampling or exponential minimum sampling to embed a watermark into the output of a language model (Kuditipudi et al., 2023). This is achieved by mapping a sequence of random numbers from a watermark key to the probabilities assigned by the language model to each possible next token in the text. The watermarked text thus encodes information about its source that can be detected by aligning the sequence of tokens with the known sequence of random numbers from the watermark key. This method is meant to be distortion-free by leveraging the inherent randomness in the language model's text generation process, ensuring that the introduction of a watermark does not change the distribution of the generated text. We use the most robust default (EXP-edit) variant of the watermark provided in the open-source implementation, with the default parameters ($n = 256$ for length of the watermark sequence, and key $= 42$ for the secret key). We note that, on the same hardware (1x NVIDIA Tesla GPU), this watermark took between 15-20x the computation time for each generation as compared to the original model or soft-watermark.

3. **Necessary and Sufficient Watermark**: The NS-Watermark can be considered an extension of soft-watermarking, and implements an efficient beam-search algorithm to control the proportion of green words in generated texts, and make the resulting watermark strength / z-score as low as possible (necessary) to guarantee a (sufficient) detectable watermark. Given the lower strength of this watermark, it aims to not significantly alter the output distribution of the base LLM. In our testing, we use the default settings in the open-source implementation of $\gamma = 0.0001$ and $\alpha = 1$.

4. **Unigram Watermark**: The Unigram Watermark is a variant of the K-gram soft-watermark with K=1, meaning the green and red lists for each token are computed using only the previous token in the sequence and resulting in a consistent green list for each new token that the model generates. This choice is made to emphasize robustness of the watermark to attacks and edits. Theoretical guarantees of the generation quality are also provided. We use the default parameters fraction $= 0.5$, strength $= 2$, a watermarking key of 0, and detection threshold of 4.0.

## 3 Experiments

### 3.1 Detectability of Watermarked Text

Upon evaluation by the GPT-judger, across all samples, the unwatermarked outputs were preferred approximately 67.4% of the time, with the watermarked outputs being preferred 25.5% of the time. The remaining samples were declared a tie by the judger. This trend held for each individual dataset as well, with the unwatermarked text completions being declared better responses for between 64.5% and 74.5% of the samples (Figure 2a). These results indicate the independent judger's ability to distinguish between unwatermarked and soft-watermarked outputs, pointing to quality degradation as a result of the watermark, or other effects of watermarking on the LLM's outputs. They are also in agreement with the winrate of stronger LLMs like GPT-4 versus smaller ones, as evaluated by GPT-based judgers in Zheng et al. (2023). To extract more insight from the results of the GPT-judger, we also examined the scores given by the judger to each output.

### 3.2 Judger Reasoning

Across the judging criteria provided in the prompt, the GPT-judger, on average, gave a higher score to the unwatermarked text completions (Figure 2b). However, the range of these scores was relatively large.

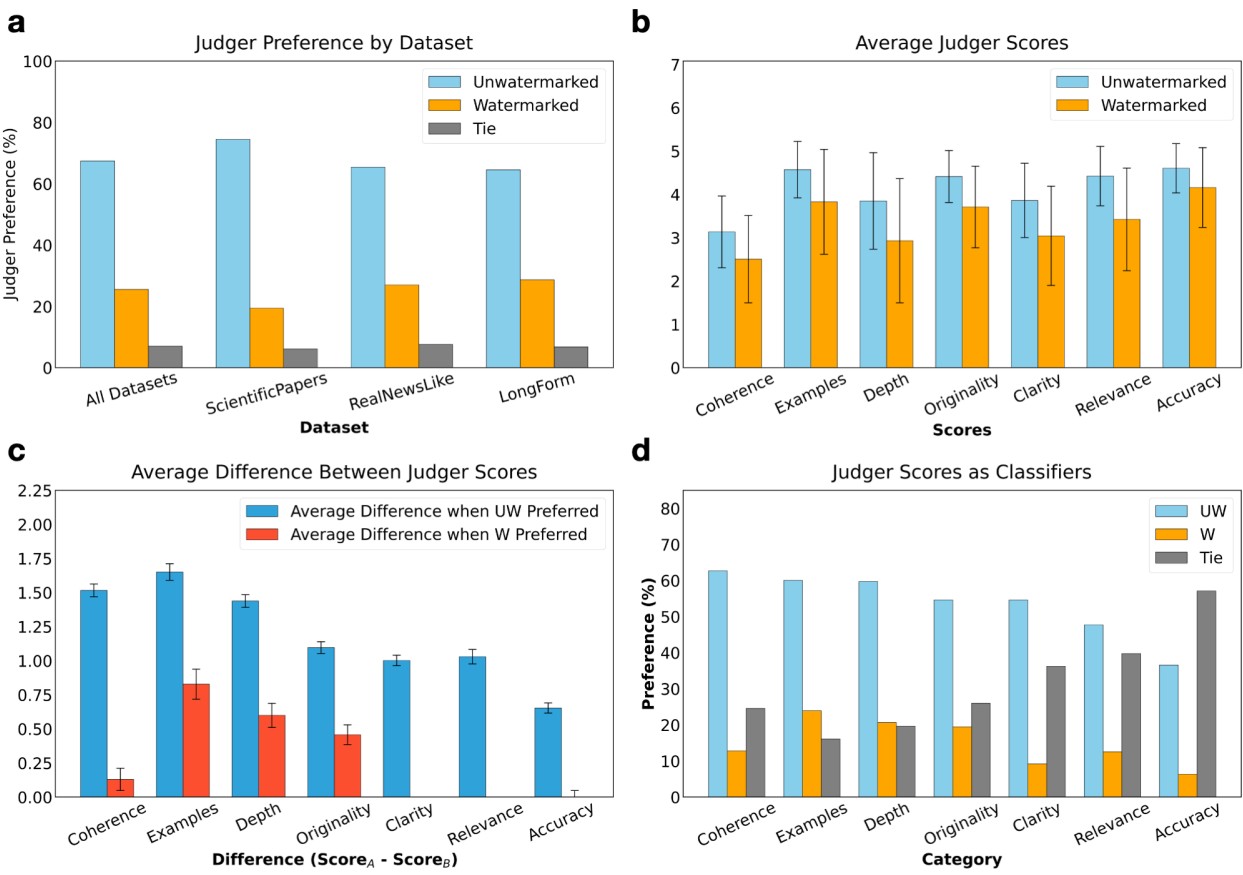

Figure 2: a) GPT-judger preference for the soft-watermark (given as the percentage of samples preferred for each class) for each dataset, separated into samples where unwatermarked outputs were preferred, watermarked samples were preferred, or neither was preferred (tie). b) Average scores for each of the 7 evaluation categories provided to the judger for all unwatermarked and watermarked samples (N = 1003). Error bars indicate the standard deviation in the scores. c) Average score differences for each evaluation category when either unwatermarked or watermarked samples were preferred. d) Judger preference when using the scores in each category as a classifier. Categories are ordered by the highest preference for unwatermarked samples as compared to watermarked samples or ties.

Seeking to determine the reasons why certain unwatermarked outputs were preferred to their watermarked counterparts, we looked at both the average difference in the scores of the two when either one was preferred, as well as used the scores in each category as a classifier. In over 50% of samples, the accuracy of both text completions was determined to be equal by the judger, indicating that the watermark does not affect this attribute of the resulting text (to the best evaluation capabilities of the judger) (Figure 2c). However, a large difference was seen in the coherence scores given to the texts, with both the average differences in the scores and judger preference for this category being significantly higher for the unwatermarked samples (Figure 2c-d). Similar results can be seen for the "use of examples" and "depth" categories, indicating that soft-watermarking tends to negatively affect these attributes of the generated text.

## 3.3 Judger Comparison

To assess the impact of judger model strength on its performance, as well as to determine the impact of the category-based ranking system in the prompt, we evaluated four different setups: Llama-2-7B-chat with the ranking-based prompt, GPT-3.5-Turbo with a simpler prompt without categorical ranking, GPT-3.5-

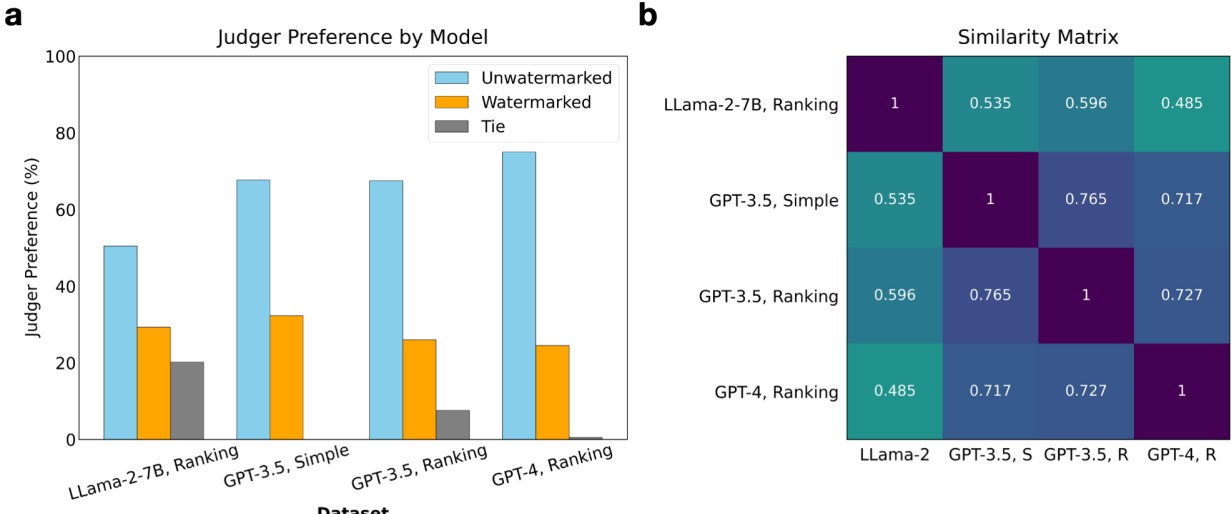

Figure 3: a) Judger preference when evaluating LLama-2-7B, GPT-3.5-Turbo, and GPT-4, with categorical ranking-based and simple prompts. All results are for a subset of N=200 samples from the RealNewsLike dataset. b) Similarities, given as the percentage of classifications that were the same, between all judgers. The models on the x-axis are abbreviated for brevity but follow the same sequence as those on the y-axis.

Turbo with the ranking prompt, and lastly, GPT-4 with the ranking prompt. Both prompts are included in Appendix B.

The results, shown in Figure 3, indicate that even a small model, Llama-2-7B, shows a preference for the unwatermarked model versus the watermarked model. It is worth noting that Llama-2 was significantly worse at adhering to the result format required by the prompt, and required repeated sampling to obtain parseable results. Upgrading to GPT-3.5-Turbo, the simpler prompt, which only required the judger to output a binary verdict for its preferred completion with no reasoning, delivered nearly identical results to the ranking-based prompt. However, the addition of the categorical ranking allowed us to extract further insights from the judger and determine which aspects of the generated text suffered the most from watermarking. GPT-4 showed an even higher preference of 75% for the unwatermarked completions, however at a substantial increase in cost. These results indicate that larger models are more discerning of the differences between unwatermarked and watermarked outputs, and may extend to newer and more capable models such as GPT-4-Turbo.

All of the GPT-based judgers agreed with each other over 71% of the time, with the highest agreement of 76.5% being between the GPT-3.5-Turbo results for the simple and categorical ranking prompts. The Llama-2-based judger agreed with the GPT-based judgers for roughly half of the samples. To further evaluate the GPT judger, we also conducted a human evaluation study where an evaluator assessed 50 randomly selected samples. Each sample consisted of the prompt and a pair of watermarked and unwatermarked generations, with a blinded human evaluator. The human evaluator preferred the unwatermarked output for 60% of the samples, agreeing with the GPT-3.5-Turbo judger 70% of the time, indicating a significant overlap in preferences between the human and GPT-based judgers.

### 3.4 Binary Classifier Performance

To further test whether watermarking discernibly altered the generated text, we employed two classification methods: a 4-layer neural network (NN) and simple logistic regression. These classifiers were trained to distinguish between watermarked and unwatermarked texts using `text-embedding-ada-002` embeddings obtained from both texts.

The neural network classifier showed promising results, yielding an average accuracy of 71% when trained on all of the datasets, along with an AUC of 0.75, false-unwatermarked rate of 30.8%, and false-watermarked rate of 30.4%. This level of accuracy was consistent, even when the model was trained on one dataset and tested on the other two, highlighting the network's ability to generalize and recognize watermarking patterns without explicit knowledge of the watermarking techniques or access to the secret key (Table 1).

Logistic regression, a simpler classification method, was also employed as an ablation study to determine if watermark detection was feasible with more basic techniques. This classifier achieved an accuracy just above random guessing, at approximately 56%, across various datasets using k-fold cross-validation with 5 folds. Notably, its highest accuracy, approximately 60%, was observed when trained on the RNL (Real-NewsLike) dataset and tested on the others, likely due to the larger sample size of 500 in the RNL dataset. Despite logistic regression's relatively lower performance compared to the MLP-based classifier, these results nevertheless suggest the presence of a watermarking signal in the texts (Table 1).

Table 1: Binary classifier (neural-network based) accuracy, AUC, false positive (FP) rate, false negative (FN) rate, and regression accuracy on each dataset for the soft-watermark. When evaluating all of the datasets together, k-fold cross-validation was used with 5-folds. For the three individual datasets, each algorithm was trained on the indicated dataset, and tested for generalizability on the other two datasets.

|  | **All Datasets** | **ScientificPapers** | **RealNewsLike** | **LongForm** |
|---|---|---|---|---|
| Accuracy | 0.711 | 0.628 | 0.649 | 0.611 |
| AUC | 0.75 | 0.665 | 0.732 | 0.724 |
| FP | 0.304 | 0.491 | 0.507 | 0.670 |
| FN | 0.308 | 0.262 | 0.200 | 0.121 |
| Regression | 0.562 | 0.573 | 0.597 | 0.515 |

## 3.5 Detectability vs. Watermarking Parameters

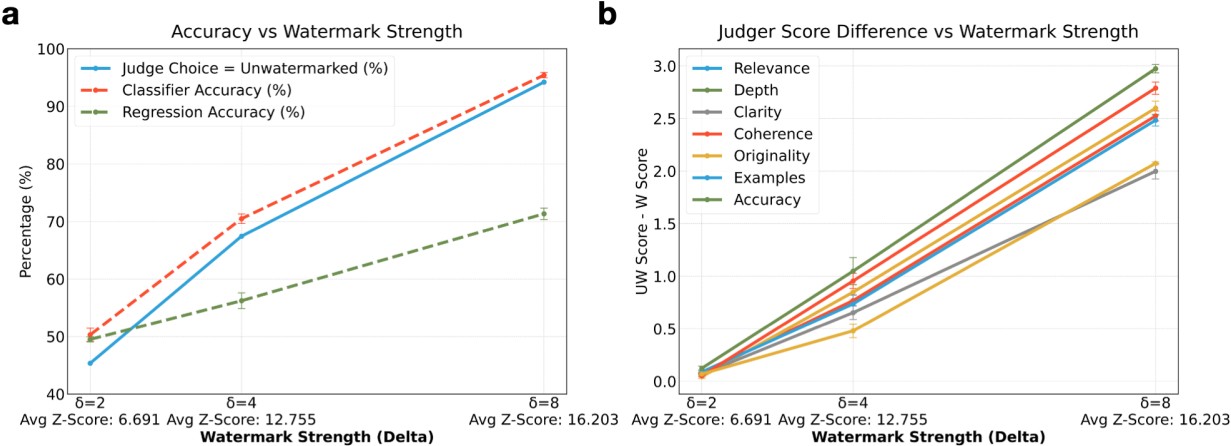

Figure 4: a) Judger preference, classifier accuracy, and regression accuracy for various soft-watermark strengths ($\delta = (2, 4, 8)$), evaluated across all of the datasets with k-fold cross-validation (5 folds). Larger $\delta$ corresponds to stronger watermarking. b) Average judger score differences for each category when varying the watermark strength. (All) error-bars represent the standard error in the measurement.

Watermarks for large language models present an inherent trade-off between robustness, computational cost, and text quality (Kirchenbauer et al., 2023a;b). We therefore evaluated the impact of the watermark strength on the accuracy of our techniques. For both of the datasets we tested, the judger's preference

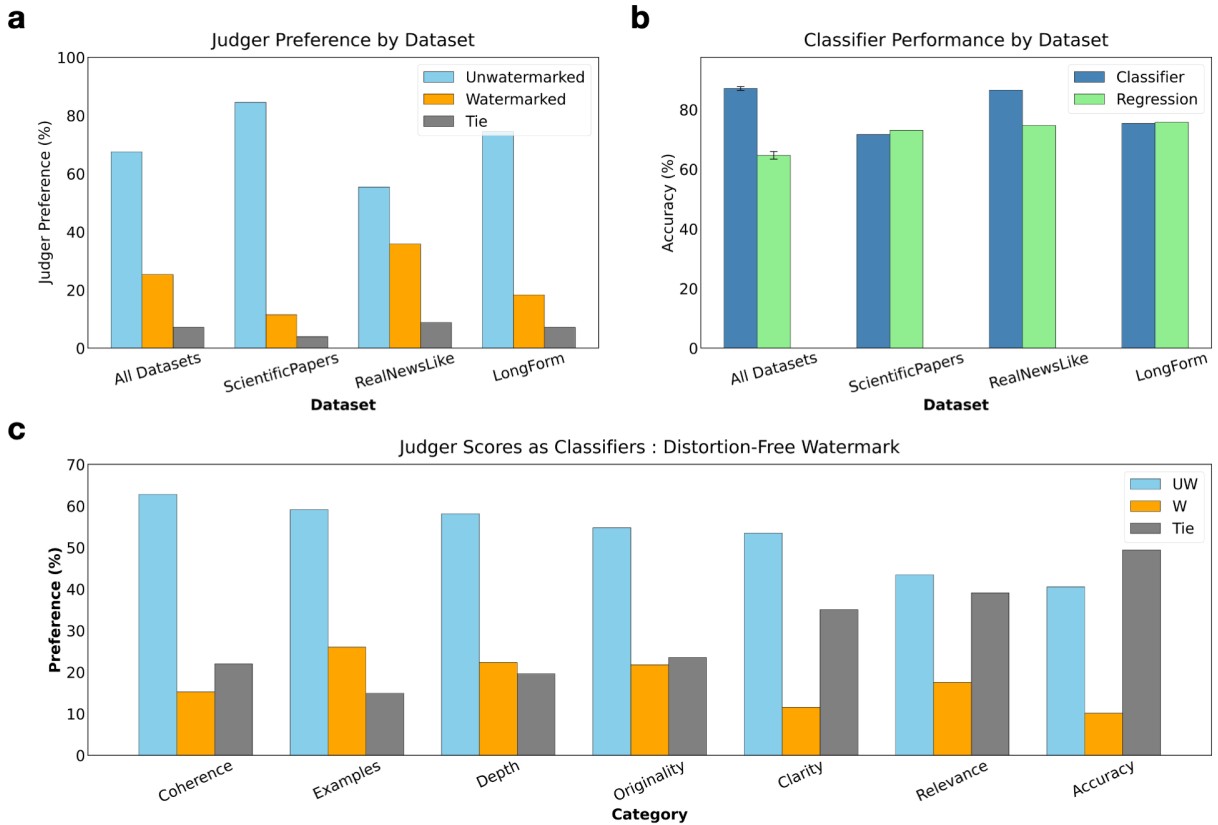

Figure 5: a) GPT-judger preference for the distortion-free watermark (given as the percentage of samples preferred for each class) for each dataset, separated into samples where unwatermarked outputs were preferred, watermarked samples were preferred, or neither was preferred (tie). b) Binary classifier (neural-network based and regression) performance on each dataset for the distortion-free watermark. k-fold cross-validation was used with 5-folds when evaluating all of the datasets. For the three individual datasets, each algorithm was trained on the indicated dataset, and tested on the other two. c) Judger preference when using the scores in each category as a classifier.

for the unwatermarked output and classifier's accuracy in predicting whether an output was watermarked increased as the watermark was made stronger (Figure 5a). At the lowest tested delta parameter of $\delta = 2$, the watermark had sufficient strength to be flagged as watermarked, but was not discernible by the judger or classifier. Additionally, as the watermarking strength was increased, the difference between the judger's scores for the unwatermarked and watermarked outputs also increased in all categories, with the largest difference being seen for the depth of the outputs (Figure 5b).

## 3.6 Other Watermarking Algorithms

Lastly, we evaluated both techniques on three other watermarking algorithms, the Robust Distortion-Free Watermark proposed by Kuditipudi et al. (Kuditipudi et al., 2023), the Unigram watermark from Zhao et al. (Zhao et al., 2023a), and the NS-Watermark by Takezawa et al. (Takezawa et al., 2023), to test the generalization of our techniques. As shown in Table 2, our results generalized across all of the techniques, with at least one method achieving over 75% accuracy for each watermark.

Taking a closer look at the robust distortion-free watermark, the judger showed a very similar preference of 67.5% for the unwatermarked samples and 25.3% for watermarked samples, with the remaining 7.2% of samples being declared ties (Figure 3.5a). On the ScientificPapers dataset, the judger preferred 84% of the

Table 2: Judger preference and classifier accuracies for each of the four watermarking algorithms we tested (soft, distortion-free, unigram, and ns-watermarks), computed over 200 samples from the RealNewsLike dataset.

| | DISTORTION-FREE | UNIGRAM | NS |
|---|---|---|---|
| GPT-Judger Preferred UW | 0.72 | 0.73 | 0.60 |
| MLP-Based Classifier | 0.87 | 0.67 | 0.57 |
| Logistic Regression Classifier | 0.65 | 0.78 | 0.76 |

unwatermarked samples, indicating an even larger degradation in quality for this class of academic text generation. Once again, the judger's decisions were most influenced by the lack in coherence, depth, and usage of examples in the watermarked text (Figure 3.5c).

Meanwhile, the MLP-based classifier achieved above 80% accuracy and 0.926 AUC in discerning watermarked samples when evaluated on all of the datasets, with linear regression also displaying a detection accuracy near 75% when trained on each individual dataset (Figure 3.5b). Collectively, these results indicate that the robust, distortion-free watermark also perceptibly affects generation quality.

## 4 Discussion

**Related Works**   In the evolving landscape of large language model applications, watermarking has emerged as a crucial technique for tracing model outputs. Most current LLM watermarking approaches involve subtly biasing the model's logits using pseudorandom distributions. These techniques vary, ranging from simple binary partitioning with 'green' and 'red' lists to more sophisticated methods. In parallel, techniques like the LLM-judger have emerged as valuable tools for comparing and benchmarking LLMs (Zheng et al., 2023), or generating annotations for instruction tuning (Wang et al., 2023b; Peng et al., 2023; Zhou et al., 2023).

In this study, we introduced two new techniques for evaluating watermarks for large language models. Fundamentally, watermarking for LLMs should remain invisible to both automated systems and human evaluators. Our study reveals that, contrary to this ideal, current watermarking techniques, including soft-watermarking and the EXP watermark, introduce detectable patterns or anomalies into the generated text. The ability of independent classifiers to detect watermarked content in LLMs without prior knowledge of the specific watermarking algorithm or secret keys is a notable finding. The effectiveness of simple classifiers like logistic regression and multi-layer perceptrons, achieving over 70% and up to 86.5% accuracy in identifying watermarked content, further underscores this point. This finding not only questions the non-detectability of these watermarks but also suggests that in practice, even methods designed to be distortion-free still suffer from degradations. These results contrast with prior work, which showed comparable results between watermarked and unwatermarked text on metrics such as perplexity (Zhao et al., 2023a) and BLEU scores (Takezawa et al., 2023). The ideal watermark should balance robustness (detectability by the intended party and resilience to attacks), and subtlety (not discernible by outside methods that don't know the watermarking key or algorithm).

The GPT-judger's scoring further adds a dimension to this understanding by highlighting the specific areas of text quality that are impacted by current watermarking methods. Watermarking, as we observed, tends to degrade text attributes like coherence and depth, whereas internal accuracy remains generally consistent. This degradation is crucial as it can compromise the utility and acceptability of watermarked texts in settings where high-quality outputs are paramount. Our results also shed light on the trade-off between watermark strength and text quality. As watermark strength increases, so does its detectability, indicating more pronounced quality degradation. With a particularly robust watermark, we see detection accuracies over 90%, all without any priors regarding the watermarking algorithm and with relatively small sample sizes used for training.

The ability to detect watermarks in texts generated using various watermarking methods suggests that subtlety might be a crucial characteristic to consider when developing these methods. Looking ahead, our

work opens new avenues for research. Future studies could employ more sophisticated classifiers, delve deeper into the nuances of how watermarking alters text generation, and ultimately use these insights to develop new watermarking techniques that implant sufficiently robust watermarks without altering the generated text perceptibly from the original model.

In conclusion, our findings provide an overview of the current state of watermarking in LLMs. The challenges in achieving undetectability and maintaining text quality are more pronounced than previously understood. As LLMs become increasingly prevalent, developing watermarking techniques that are both robust and subtle is crucial.

**Code Availability:** The source code for all experiments is available at `https://github.com/su-karanps/watermark_eval`.

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

# A  Appendix A: Watermarking Algorithms

**Watermark Strength:**

Table 3: True and false positive rates for each of the four watermarking algorithms we tested (soft, distortion-free, unigram, and ns-watermarks), computed over 200 samples from the RealNewsLike dataset.

|                     | SOFT  | DISTORTION-FREE | UNIGRAM | NS   |
|---------------------|-------|-----------------|---------|------|
| False Positive Rate | 0.015 | 0               | 0.005   | 0.01 |
| True Positive Rate  | 0.96  | 0.67            | 0.535   | 1.0  |

**Example Response Set:**

Table 4: Example response set for the soft and distortion-free watermarks. Incoherent sections of the watermarked outputs that likely contribute to the judger's decision are highlighted. For longer such sections, only the first sentence is highlighted for visibility. DF-Watermark refers to the distortion-free watermark.

| PROMPT | UNWATERMARKED | SOFT-WATERMARK | DF-WATERMARK |
|---|---|---|---|
| A little over 5 acres of land in a prime area of Western North Carolina. Upon arrival to the home site, you'll pass gorgeous mountain views, a babbling creek and pasture land. This property has a storage shed already in place and the land is ready for your use which | makes it easy to start building your dream home. The property is located in a quiet and peaceful area with a short distance to the Blue Ridge Parkway and all the outdoor activities it has to offer. The nearby town of Spruce Pine offers shopping, dining and entertainment options. Don't miss out on this amazing opportunity to own a little piece of heaven.

Directions: From the intersection of Hwy 221 and Hwy 70 in Spruce Pine, take Hwy 70 East for approximately 4 miles. Property will be on the right just past the Blue Ridge Parkway Overlook. Look for the sign. | could be a homestead, farm, or a place to park your RV. Close to the Blue Ridge parkway, Boone and Watauga river making this a prime location for outdoor enthusiasts or just a great place to unwind. Property is being sold as is with no disclosures or warranties. Seller would like a 30 day inspection period, and a cash deal would be the most likely outcome, however would discuss with the right offer. Agent is related to the seller. This is a great chance to live the country life close to everything. | includes double rural zoning. With a little creativity and some elbow grease this property can be turned into your very own piece of paradise. Sunset views are nothing short of breathtaking. Location is just minutes from the nearby cities of Marshall, Bryson City and Hayesville. This property is priced to sell and won't last long. Call today and schedule a viewing! When we first meet Scott Morrison, he is sitting in his office. He is wearing a sky blue shirt with a hint of a smile, looking worn and exhausted. He has been awake for days...

Scott Morrison is the 30th Prime Minister of Australia, serving since August 2018. Since his appointment, he has been at the forefront of national issues, from climate change to the economy..., and dealing with the fallout of the Kavenaugh affair. |

# B   Appendix B: Judger

**Judger Prompt and Example Output:**

Table 5: Judger prompt and completion from GPT-3.5-Turbo for the example response set given in Table 4. In this example, the randomization resulted in LLM A being the watermarked completion, and LLM B being the unwatermarked completion.

| PROMPT | JUDGER RESPONSE |
|---|---|
| Please act as an impartial judge and evaluate the quality of the text completions provided by two large language models to the prompt displayed below. Assess each response according to the criteria outlined, using a 1-5 Likert scale. After scoring each criterion, provide a summary for each response, including examples that influenced your scoring. Additionally, ensure that the order in which the responses are presented does not affect your decision. | In my assessment, the superior response is from LLM B. |

**Criteria:**

1.  Relevance to the prompt

2.  Depth of detail

3.  Clarity of writing

4.  Coherence and logical flow

5.  Originality and insight

6.  Use of specific examples

7.  Accuracy of information

Start with a brief statement about which response you think is superior. Then, for each response and criterion, provide a score, followed by a brief justification for that score. At the very end of your response, declare your verdict by choosing one of the choices below, strictly following the given format:

"[[A]]: [list of scores]" if assistant A is better,
"[[B]]: [list of scores]" if assistant B is better, or
"[[C]]" for a tie.

[Prompt]

[LLM A's Answer]

[LLM B's Answer]

1. Relevance to the prompt:
- LLM A: 3
- LLM B: 5

LLM B's response directly addresses the prompt by emphasizing the opportunity to build a dream home and providing directions to the property. In contrast, LLM A's response focuses more on general information about the location and potential uses of the land, which is somewhat relevant but not as specific.

2. Depth of detail:
- LLM A: 4
- LLM B: 3

LLM A provides more details about the property, such as mentioning the presence of mountain views, a babbling creek, pasture land, and a storage shed. On the other hand, LLM B's response lacks specific details about the land itself, instead focusing more on the nearby town and activities.

.
.
.

7. Accuracy of information:
- LLM A: 5
- LLM B: 5

Both responses provide information that does not require fact-checking. They introduce no factual inconsistencies.

Based on the evaluation above, my verdict is:

[[B]]: 3, 3, 5, 5, 4, 3, 5

Table 6: Simple judger prompt and completion from GPT-3.5-Turbo for the example response set given in Table 4. In this example, the randomization resulted in LLM A being the unwatermarked completion, and LLM B being the watermarked completion.

| SIMPLE PROMPT | JUDGER RESPONSE |
| --- | --- |
| Please act as an impartial judge and evaluate the quality of the text completions provided by two large language models to the prompt displayed below. You should choose the LLM that completes the prompt better. Begin your evaluation by comparing the two responses and provide a short explanation. Avoid any position biases and ensure that the order in which the responses were presented does not influence your decision. Do not allow the length of the responses to influence your evaluation. Be as objective as possible. After providing your explanation, output your final verdict by strictly following this format: "[[A]]" if assistant A is better, "[[B]]" if assistant B is better, and "[[C]]" for a tie. [Prompt]

[LLM A's Answer]

[LLM B's Answer] | The response provided by LLM A provides more specific information about the property and its potential uses. It mentions that the land could be used as a homestead, farm, or a place to park an RV. It also highlights the proximity to the Blue Ridge Parkway, Boone, and Watauga River, appealing to outdoor enthusiasts. Additionally, it mentions that the property is being sold as is with no disclosures or warranties, and that a cash deal is the most likely outcome.

On the other hand, LLM B's response is focused more on the location and nearby amenities. It mentions that the property is located in a quiet and peaceful area, with nearby access to the Blue Ridge Parkway and various outdoor activities. It also briefly mentions the nearby town of Spruce Pine and the options for shopping, dining, and entertainment. The response concludes with directions to the property from a specific intersection.

Overall, both responses provide relevant information, but LLM A's response provides more details about the property and its potential uses. Therefore, based on the evaluation, [[A]] is better. |

## C    Appendix C: Binary Classifier

The classifier is a multi-layer perceptron (MLP) with the following architecture:

- Input size: 1536

- Output size: 1

- Hidden layers: 512, 256, 128, 64

- Activation: ReLU between layers, sigmoid at the output

Optimization was performed using the Adam optimizer with: $\beta_1 = 0.5$, $\beta_2 = 0.999$, learning rate and weight decay determined through a hyperparameter grid search (parameters shown in Table C). A dynamic learning rate scheduler was used (`torch.optim.lr_scheduler.ReduceLROnPlateau`) with a factor of 0.5 and a patience of 50 epochs. Training was conducted for 150 epochs.

| Hyperparameter | Value |
|---|---|
| Adam Weight Decay | Grid search over $\{2e^{-4}, 2e^{-3}, 2e^{-2}\}$ |
| Learning Rate | Grid search over $\{2e^{-5}, 2e^{-4}, 2e^{-3}\}$ |
| Batch Size | Grid search over $\{50, 75, 100\}$ |
| Dataset Randomization | Grid search over $\{$On, Off$\}$ |

Table 7: Binary classifier hyperparameters

