# OpenReview forum: "New Evaluation Metrics Capture Quality Degradation due to LLM Watermarking"
_TMLR — Accepted by TMLR_

### Review · Reviewer_CBc5 · 2024-01-10

**Summary Of Contributions:**

To evaluate watermarked texts generated by LLMs, the authors provide two methods. One is automated judgment with GPT: By using carefully designed prompts, we let GPT evaluate text quality from several factors with the Likert scale, such as coherence, use of examples, depth of detail, originality, clarity, relevance to the prompt, and accuracy of information. The other is detection by binary classifier with multi-layer neural networks or simple logistic regression built on top of text-embeding-ada-002. In the experiments, they found that both the GPT-judgers and binary classifiers can detect watermarked texts with around 70% accuracy on average. This detectability is enhanced with more intense watermarks. In the evaluation by the GPT-judgers, they specifically found that watermarked and unwatermarked texts obtained large score gaps for coherence, use of examples, depth of details, and originality, while there were no significant score gaps for clarity, relevance, and accuracy.

**Audience:**

Yes

**Broader Impact Concerns:**

LLM watermarking is undoubtedly a vital technique to prevent false information and plagiarism from machine-generating texts. The authors discussed this perspective slightly in the introduction.

**Claims And Evidence:**

Yes

**Requested Changes:**

## Major comments

As I discussed above, the potential suggestion is to add a few more baseline watermarks (in order to see the robustness of the proposed evaluation methods).

Additionally, I came up with several spontaneous ideas to extend the content of this paper. You can have a look and incorporate them into the paper.

- Ensemble classifier for watermark detection: Though the GPT-judger and neural network classifiers are not powerful enough, as I criticized above, it makes sense to regard them as weak learners and apply ensemble methods or boosting. It must be interesting to see how powerful the detection accuracy will be.
- Ensemble score with the GPT-judger: Currently, the GPT-judger only adopts a single LLM as a judger and lets it output the Likert scales. We can probably use multiple LLMs and average their Likert scales. Such an ensemble method may provide a more stable evaluation.

## Minor comments

- In Section 2.1 (the fourth line of "1. Automated Judgement with GPT"), should "GPT-4" be "GPT-3.5-Turbo"?
- For typesetting double quotation marks, use `` ... '', that is, two backquotes and two single quotes.
- In Section 3.1, the term "GPT-judger" appears for the first time. I suppose this indicates the automated judgment with GPT introduced in Section 2.1. It is better to introduce the terminology formally.
- In Section 3.4, it is better to summarize the AUC scores and so on in a table. Currently, those watermark detection performances scatter here and there.
- In Section 3.5 (the fourth last line), it seems strange to say "the watermark was still robust enough to be detected" because the detection accuracy is only around 50% (chance rate) with $\\delta=2$.

**Strengths And Weaknesses:**

## Strengths

- Novelty: This work seems to investigate the possibility of LLM-as-a-judge for the quality of watermarked texts as far as I surveyed.
- Proposing factors to characterize text quality: In this paper, seven factors (coherence, clarity, accuracy, ...) to characterize text quality are proposed. These factors are reasonable, and surprisingly, we see significant score differences (between the cases when watermarked and unwatermarked texts are preferred) only for some factors, such as coherence and originality (Figure 2c). This may partially describe the characteristics of some watermarking methods.
- Simplicity of proposed methods: Both the GPT-judger and binary classification are very simple and not costly to use in practice.

## Weaknesses

- Limited numbers of baseline watermarks: The authors evaluated only two watermark methods, Soft-Watermark and Distortion-Free Watermark. Though they discuss why they chose these two methods to evaluate (and perhaps mainly because of the availability of public implementation), some other works (cited in this paper) publicize their implementation. For instance, [Takezawa et al. (2023)](https://github.com/yukiTakezawa/necessary_and_sufficient_watermark) (not included in their paper, though) and [Zhao et al. (2023a)](https://github.com/XuandongZhao/Unigram-Watermark) provide their codebases. Evaluation using these other methods is important to assess the robustness of the proposed evaluation methods (i.e., how universally we observe the tendency in Fig. 2) across different watermarking methods.
- Insufficient detection accuracy: With both GPT-judgers and binary classifiers, the detection performances of watermarked texts remain up to 70% accuracy (unless we increase the watermark intensity $\\delta$ of the Soft-Watermark). This observation is interesting because 70% accuracy cannot be regarded as practically reliable enough, and hence the existing watermarking methods partially succeed in being undetectable. For this reason, I don't think we can confidently claim that "current watermarking methods are detectable by even simple classifiers" (in the abstract) or "the presence of a detectable watermarking signal in the texts" (in the end of Section 3.4).

---

> ### Author Response · Authors · 2024-03-02
> **Response by Authors for Reviewer CBc5**
>
> We sincerely thank the reviewer for their thorough and insightful suggestions to our paper. We address each comment individually below:
>
> > Limited numbers of baseline watermarks: The authors evaluated only two watermark methods, Soft-Watermark and Distortion-Free Watermark… For instance, Takezawa et al. (2023) (not included in their paper, though) and Zhao et al. (2023a) provide their codebases. Evaluation using these other methods is important to assess the robustness of the proposed evaluation methods (i.e., how universally we observe the tendency in Fig. 2) across different watermarking methods.
>
> * We have added evaluation results using our method for the Unigram Watermark (Zhao et al. 2023), where we see a similar judger preference of 73% for the unwatermarked outputs, and the NS-Watermark (Takezawa et al. 2023), where we see a preference of 60%. Classifier accuracy for both watermarks exceeds 75%. We have updated the manuscript with these results in Section 3.6.
>
> > Insufficient detection accuracy: With both GPT-judgers and binary classifiers, the detection performances of watermarked texts remain up to 70% accuracy (unless we increase the watermark intensity of the Soft-Watermark). This observation is interesting because 70% accuracy cannot be regarded as practically reliable enough, and hence the existing watermarking methods partially succeed in being undetectable. For this reason, I don't think we can confidently claim that "current watermarking methods are detectable by even simple classifiers" (in the abstract) or "the presence of a detectable watermarking signal in the texts" (in the end of Section 3.4).
>
> * Thank you for your insightful feedback on our manuscript. In revising our manuscript, we've first sought to contextualize the detection accuracy of our watermarked texts in relation to the variability observed in the performance of different LLMs. Drawing on the winrate metrics from Chatbot-Arena [1], we believe that the ~70% detection accuracy achieved in our study is notable, reflecting a similar range of performance variability among LLMs. However, we agree that current watermarking methods do achieve partial success in remaining undetected. Given this, we’ve refined the language in our abstract and Section 3.4 to more closely align our claims with the observed performance of the GPT-judgers and binary classifiers.
>
> * Regarding the suggestion to employ ensemble methods or boosting, we explored various approaches but found that these techniques did not outperform the strongest single model in our experiments. We also considered the potential of averaging Likert scales across the multiple LLMs that we evaluated in our comparison study between the LLM-judgers. However, our preliminary efforts did not yield substantial improvements, possibly due to the inherent variability in the responses of the different LLMs.
>
> > In Section 2.1 (the fourth line of "1. Automated Judgement with GPT"), should "GPT-4" be "GPT-3.5-Turbo"?
>
> * We apologize for the confusion. To assess the impact of the LLM used, we did a comparison study that included GPT-4. However, for the main analyses conducted in the remainder of the paper, we opted for GPT-3.5-Turbo due to its balance of performance and accessibility. We have clarified this in Section 2.1.
>
> > For typesetting double quotation marks, use `` ... '', that is, two backquotes and two single quotes.
>
> * Thank you for pointing this out. We have fixed all occurrences.
>
> > In Section 3.1, the term "GPT-judger" appears for the first time. I suppose this indicates the automated judgment with GPT introduced in Section 2.1. It is better to introduce the terminology formally.
>
> * Thank you for the good suggestion! We have modified the introduction of this method in Section 2.1 to include the term.
>
> > In Section 3.5 (the fourth last line), it seems strange to say "the watermark was still robust enough to be detected" because the detection accuracy is only around 50% (chance rate) with delta=2.
>
> * Apologies for the misunderstanding. We meant to highlight that the watermark had sufficient strength at δ=2 to be flagged as watermarked, yet was not detectable by our evaluation methods. We have updated the manuscript to make this more clear.
>
> > In Section 3.4, it is better to summarize the AUC scores and so on in a table. Currently, those watermark detection performances scatter here and there.
>
> * Thank you for this suggestion! We have replaced Figure 4 with a table summarizing the accuracy, AUC, false positive, and false negative rates for each method.
>
> Thank you again for your valuable feedback, which has enabled us to improve the paper! Please let us know if you have any further questions and we are happy to follow up.
>
> References:
>
> [1] Zheng et al., Judging LLM-as-a-Judge with MT-Bench and Chatbot Arena, NeurIPS 2023

---

### Review · Reviewer_41vW · 2024-01-15

**Summary Of Contributions:**

This paper proposed two evaluation methods for watermarking methods. First, this paper proposed Automated Judgement with GPT to evaluate how text quality degrades by watermarking methods and then evaluated the existing watermarking methods by using the proposed methods. This paper also proposed a method to evaluate the undetectability of watermarking methods by using supervised learning. Then, this paper demonstrated that the generated texts with existing watermarks are distinguishable from unwatermarked texts with high accuracy.

**Audience:**

Yes

**Claims And Evidence:**

Yes

**Requested Changes:**

* Could you please make the code public? Making the code public is very important for reproducibility, especially for the evaluation metric.
* Sec. 2.3 explained the hyperparameter setting for each method, but does not report the false positive rates of watermarking methods. These values are very important to measure the strength of watermarks. Could you please report the false positive rate in Sec. 2.3?
* Please see other comments listed in the weakness section.

**Strengths And Weaknesses:**

## Strength
* This paper is well-written and easy to read.
* This paper proposed two evaluation methods, and the experimental results are solid. Regarding the undetectability of watermarking methods, this paper demonstrated that even the DF-Watermark, which has the distortion-free property, is distinguishable from unwatermarked texts.

## Weakness
* This paper claimed that the existing evaluation metrics for watermarking methods, e.g., perplexity, can not capture the semantic coherence or contextual relevance of generated texts and then proposed the evaluation methods using GPT. However, the reviewer feels that this advantage of the proposed method over the existing evaluation metric is unclear. For instance, how do the results of evaluation by the proposed method, Automated Judgement with GPT, differ from the results by perplexity and other existing evaluation metrics? A similar result that the insertion of watermarks degrades the text quality has already been shown in a prior study using perplexity [1]. It would be better for the paper to clearly describe when the proposed evaluation method should be used instead of existing evaluation metrics. Furthermore, it would be better to discuss what different experimental results can be obtained using the proposed evaluation method.

* This paper focused on the evaluation metrics for watermarking methods and proposed the new evaluation metric, Automated Judgement with GPT. However, it is a very fundamental topic in NLP to measure the correctness of generated texts, and there exist a lot of existing methods, including perplexity, BLUE score, and LLM-based methods (e.g., [2]). Are there any issues that are specific to evaluating generated texts with watermarks? The reviewer did not understand the motivation to propose a new evaluation metric to assess text quality degradation caused by watermarking.

## Reference
[1] Kirchenbauer et al., A Watermark for Large Language Models, ICML 2023

[2] Zheng et al., Judging LLM-as-a-Judge with MT-Bench and Chatbot Arena, NeurIPS 2023

---

> ### Author Response · Authors · 2024-03-02
> **Response by Authors for Reviewer 41vW**
>
> We truly appreciate the reviewer for their valuable comments. We address the concerns of the review below:
>
> > Report the false positive rates of the watermarking methods. These values are very important to measure the strength of watermarks.
>
> * Thank you for this good suggestion! We have updated the Appendix with both the true and false positive rates that we observed in our experiments for all of the tested watermarks (extended to four watermarks). Across the board, we observe very low false positive rates, with the true positive rate being close to 1.0 for the soft and NS-watermarks and slightly lower for the other two (0.67 for the distortion-free watermark and 0.54 for the unigram watermark). These values indicate relativity strong watermarks, and the detection accuracy of our proposed methods would likely increase if the strength was increased further, beyond the default parameters that we tested.
>
> > Could you please make the code public? Making the code public is very important for reproducibility, especially for the evaluation metric.
>
> * Thank you for this suggestion. We have uploaded implementations of our evaluation metrics to an anonymized repository, and updated the paper with a link.
>
> > This paper claimed that the existing evaluation metrics for watermarking methods, e.g., perplexity, can not capture the semantic coherence or contextual relevance of generated texts and then proposed the evaluation methods using GPT. However, the reviewer feels that this advantage of the proposed method over the existing evaluation metric is unclear. For instance, how do the results of evaluation by the proposed method, Automated Judgement with GPT, differ from the results by perplexity and other existing evaluation metrics? A similar result that the insertion of watermarks degrades the text quality has already been shown in a prior study using perplexity [1]. It would be better for the paper to clearly describe when the proposed evaluation method should be used instead of existing evaluation metrics. Furthermore, it would be better to discuss what different experimental results can be obtained using the proposed evaluation method.
>
> * Thank you for bringing up this point. To the best of our knowledge, the results showing quality degradation through perplexity are not conclusive across the watermarks we tested (note that we added two watermarking algorithms to our analysis in the revised manuscript). For instance, the Unigram watermark produces perplexity values comparable to those of human-generated and unwatermarked text with Llama-7B. However, we still observe quality degradation through our methods.
>
> * We have made this point more clear in our updated manuscript, along with expanding our discussion of the potential applications of our method over others.
>
> > This paper focused on the evaluation metrics for watermarking methods and proposed the new evaluation metric, Automated Judgement with GPT. However, it is a very fundamental topic in NLP to measure the correctness of generated texts, and there exist a lot of existing methods, including perplexity, BLUE score, and LLM-based methods (e.g., [2]). Are there any issues that are specific to evaluating generated texts with watermarks? The reviewer did not understand the motivation to propose a new evaluation metric to assess text quality degradation caused by watermarking.
>
> * Thank you for this comment. Similarly to perplexity, metrics like BLEU scores can fail to accurately adjudicate the relative correctness of generated texts in the context of watermarking. As an example, BLEU scores for text generated using the NS-watermark are very similar to those from text generated without the watermark, and yet an MLP trained on text embeddings from both texts is able to correctly separate watermarked samples with an accuracy of over 75%.
>
> * Our proposed metrics are designed to evaluate the relative impact of watermarking on text quality and coherence, addressing a gap not covered by existing NLP metrics. This allows for a more nuanced and fine-tuned assessment of watermarking techniques, ensuring that the text remains both authentic and undistorted. The insights from our GPT-judger also reveal the specific attributes (e.g. coherence, use of examples, etc.) of the text that are most altered by watermarking, providing points for improvement for future algorithms.
>
> * We have emphasized these points in the updated manuscript to provide a more clear motivation for the work.
>
> Thank you again for your helpful feedback! Please let us know if you have any further questions and we are happy to follow up.
>
> References:
>
> [1] Zhao et al., Provable Robust Watermarking for AI-Generated Text, ICLR 2024
>
> [2] Takezawa et al., Necessary and Sufficient Watermark for Large Language Models, arXiv, 2023

---

### Review · Reviewer_Srd4 · 2024-02-26

**Summary Of Contributions:**

This paper proposes two easy-to-use methods to evaluate existing watermarking methods. It argues that existing watermarks can be easily detected and lower the quality of the original text. The total content is clear, and the methods are introduced well. However, some key issues are identified in this paper, which might need to be fixed before the final acceptance.

**Audience:**

No

**Claims And Evidence:**

Yes

**Requested Changes:**

Weaknesses:

The main issue of this paper is the ignore of the significance of existing watermarking methods. Lowering the quality is easy to understand. However, easy-to-detect might not be a drawback? The logic chain here is broken, and literature is used to force readers to believe this claim. In my opinion, a new section should be added to convince readers to believe that the two issues are significant in the field of watermarking. By doing so, it can also increase the significance of this paper in the field.

**Strengths And Weaknesses:**

Strength:

1) This paper focuses on a very important problem. Watermarking is a very important technique to enable tracing the IP of AI models. Exploring something regarding watermarking is always important at the current era.

2) This paper demonstrates findings clearly, readers can easily understand the proposed two methods.

Weaknesses:

The main issue of this paper is the ignore of the significance of existing watermarking methods. Lowering the quality is easy to understand. However, easy-to-detect might not be a drawback? The logic chain here is broken, and literature is used to force readers to believe this claim. In my opinion, a new section should be added to convince readers to believe that the two issues are significant in the field of watermarking. By doing so, it can also increase the significance of this paper in the field.

---

> ### Author Response · Authors · 2024-03-02
> **Response By Authors for Reviewer Srd4**
>
> We are grateful to the reviewer for their constructive feedback. Our paper offers two new automated approaches to evaluate LLM watermarking: 1) quality/preference evaluation using a critic LLM, and 2) detectability using a classifier on embeddings. The first approach shows that several popular watermarking methods can reduce the quality of the generation, which is a useful insight as you noted.
>
> Our second detectability measure is motivated by recent watermarking methods that aim to minimize distortion in the generated text [1]. Low distortion is desirable because it shows that watermarking does not introduce significant side effects on the generation. Detectability is a natural way to quantify distortion; the more distorted the generation is compared to unwatermarked text, the easier it would be for a classifier to detect. We have clarified this motivation in the text.
>
> We would also like to highlight the difference between detectability of the watermark by the intended, knowledgeable party using the proper algorithm, and detectability by an uninformed third party with no knowledge of the watermarking secret key or algorithm (which could undermine the watermark’s subtlety and effectiveness). While the former is certainly a positive feature, and implanting a robust watermark is a crucial issue in the field, the latter suggests a watermark that is not subtle enough and therefore contrasts with the goal of effective watermarking with minimal side effects.
>
> Per the requested change, we have added sections to the introduction and discussion making this distinction, and further convincing readers that both robustness and subtlety are important factors to consider.
>
> Thank you again for your helpful feedback, which has enabled us to strengthen the paper! Please let us know if you have any further questions and we are happy to follow up.
>
> References:
>
> [1] Kuditipudi et al., Robust Distortion-free Watermarks for Language Models, arXiv 2023

---

### Author Response · Authors · 2024-03-03
**Thank You**

Dear Reviewers and Action Editor,

We would like to express our sincere gratitude for the valuable feedback and constructive criticisms provided on our manuscript. We have carefully updated the paper to incorporate all of the reviewers’ suggestions. To facilitate an easier review of the changes made, we have highlighted all major additions and revisions in green within the manuscript.

Thank you again for your time and effort in reviewing our manuscript.

The Authors

---

### Decision · Action_Editor_1CvG · 2024-04-05

**Recommendation:** Accept as is

**Comment:**

The paper introduces two evaluation methods for assessing watermarked texts: Automated Judgment with GPT and binary classifier detection. Using GPT prompts, text quality factors like coherence, originality, and accuracy are assessed. Binary classifiers, leveraging text embeddings, detect watermarks effectively. Results show both methods achieve around 70% accuracy. This detectability is enhanced with more powerful watermarks.

The LLM watermarking is an emerging topic, with numerous methods being developed. The proposed approach is both intriguing and practical. All reviewers have recommended acceptance, and I also vote for acceptance.

**Audience:**

LLM watermarking is an emerging technique and important research topic. Thus, the paper will attract many researchers who are working on LLM.

**Claims And Evidence:**

Yes.